# Determinants of life expectancy in most polluted countries: Exploring the effect of environmental degradation

**Mohammad Mafizur Rahman**[1,2]*, **Rezwanul Rana**[3], **Rasheda Khanam**[1,2]

**1** School of Business, University of Southern Queensland, Toowoomba, Queensland, Australia, **2** Centre for Health Research, University of Southern Queensland, Toowoomba, Queensland, Australia, **3** Centre for the Health Economy, Macquarie Business School, Macquarie University, Sydney, NSW, Australia

\* mafiz.rahman@us.edu.au

## Abstract

### Background

Better understanding of the determinants of national life expectancy is crucial for economic development, as a healthy nation is a prerequisite for a wealthy nation. Many socioeconomic, nutritional, lifestyle, genetic and environmental factors can influence a nation's health and longevity. Environmental degradation is one of the critical determinants of life expectancy, which is still under-researched, as the literature suggests.

### Objectives

This study aims to investigate the determinants of life expectancy in 31 world's most polluted countries with particular attention on environmental degradation using the World Bank annual data and British Petroleum data over the period of 18 years (2000–2017).

### Methods

The empirical investigation is based on the model of Preston Curve, where panel corrected standard errors (PCSE) and feasible general least square (FGLS) estimates are employed to explore the long-run effects. Pairwise Granger causality test is also used to have short-run causality among the variables of interest, taking into account the cross-sectional dependence test and other essential diagnostic tests.

### Results

The results confirm the existence of the Preston Curve, implying the positive effect of economic growth on life expectancy. Environmental degradation is found as a threat while health expenditure, clean water and improved sanitation affect the life expectancy positively in the sample countries. The causality test results reveal one-way causality from carbon emissions to life expectancy and bidirectional causalities between drinking water and life expectancy and sanitation and life expectancy.

**Data Availability Statement:** All data are publicly and freely available in the World Development Indicators published by World Bank and BP statistical Review of World Energy.

**Funding:** The authors received no specific funding for this work.

**Competing interests:** The authors have declared that no competing interests exist.

## Conclusion

Our results reveal that environmental degradation is a threat to having improved life expectancy in our sample countries. Based on the results of this study, we recommend that: (1) policy marker of these countries should adopt policies that will reduce carbon emissions and thus will improve public health and productivity; (2) environment-friendly technologies and resources, such as renewable energy, should be used in the production process; (3) healthcare expenditure on a national budget should be increased; and (4) clean drinking water and basic sanitation facilities must be ensured for all people.

## 1. Introduction

Numerous recent studies labelled environmental degradation as the most critical determinant of life expectancy in the world today. Following Adams and Klobodu [1] and Mohsin, Abbas [2], this study has used CO2 emission levels to measure environmental degradation. According to the World Health Organization [3], 4.2 million premature deaths in the world in 2016 were caused by ambient air pollution, and this is projected to increase further as 9 out of 10 of the world's population resides in places with hazardous air quality [4]. Environmental degradation can adversely impact population health in several ways. Severe outdoor air pollution is responsible for rising chronic diseases (e.g. Asthma, heart diseases and lung cancer) [5, 6] and increasing premature mortality [7]. Others concluded that environmental degradation increases the likelihood of waterborne diseases [8] such as malaria and dengue fever [9, 10]. Previous studies also concluded that environmental degradation increases the variability in the ecosystem, increasing the probability of floods and droughts [11]. As a result, environmental degradation might cause adverse variations in food production and water quality, which contributes to higher mortality, particularly among infant and elderly populations, as well as vulnerable people from lower socioeconomic background. Wen and Gu [12] and Wang et al. [13] found that air quality critically impacts the longevity of the elderly population who has minimal ability to cope with environmental degradation due to other comorbidities. Similarly, Majeed and Ozturk [14] demonstrated that countries with a higher level of environmental degradation experience greater infant mortality and vice-versa.

Despite the above empirical evidence, many developing countries continue to disregard decisive actions against environmental degradation. Chasing higher economic growth, these developing countries exert a lot of pressure on environmental resources (e.g. water, land and forest), and their increasing production fosters higher CO2 emissions and industrial wastes [15–18]. Countries with high levels of environmental degradation fail to realize the long-run positive impact of strong environmental law on economic growth and health [19]. Their lack of focus on the environment warrants further considerations. No study so far has examined the determinants of life expectancy in most polluted countries with due attention to the adverse effect of environmental degradation on population's longevity. This motivates us to pursue this research to fill up the current research gap.

This paper used life expectancy as a public health outcome, and the objective of this research is to examine the key determinants of life expectancy in the most polluted countries of the world. Our main variables of interest are economic growth, proxied by GDP per capita and environmental degradation, proxied by $CO_2$ emissions per capita. Other controlled/explanatory variables are health expenditure per capita, access to essential drinking water and

sanitation services. The rationale for selecting these 31 most polluted countries is all of these countries are developing countries where average life expectancy is lower (70 years) compared to that of developed countries (around 80 years). The justification for selecting other explanatory variables in this study are: average per capita $CO_2$ emissions are six metric tons in these sample countries; average per capita health expenditure is lower (US$700) compared to high income (US$ 5,600) and OECD (US$ 5,041) countries; still, 15% of the population have no basic drinking water service; and, 29% of the population do not use basic sanitation facilities [20]. Moreover, the variables used in this paper are along the line of past literature.

The primary hypothesis of the study is that the positive correlation between economic growth and life expectancy will persist, and environmental degradation will have a significantly higher negative impact on life expectancy than often estimated in empirical studies. Hence, the aim is to measure the validity of Preston's curve and the impacts of $CO_2$ emission on longevity. Another hypothesis is that health expenditure per capita [21–23], availability of safe drinking water and sanitation facilitates [24–26] will positively influence longevity. Following the studies of Majeed and Ozturk [14], Ebenstein et al. [27] and Mohmmed et al. [28], $CO_2$ emission is used as a measure of environmental pollution.

The main contributions of this research to the existing literature can be noted as follows: (i) the paper has used longitudinal data to determine the factors impacting life expectancy, and longitudinal data provide multiple observations for each item which facilitates reliable research method, eliminates estimation bias and reduces the problem of multicollinearity [29]; (ii) the study has also used appropriate diagnostic tests to check the accuracy of the model; (iii) to the best of knowledge of the authors, this is the first study of its kind that used long-term data to estimate the determinants of life expectancy in the world's most polluted countries; (iv) the findings of health outcomes at the individual country-level revealed by clinical and epidemiological studies are seldom used for macroeconomic policy implications [30]; this study addressed this issue. Our findings will be critically important to implement effective public health and environmental policies, in particular with an increasing number of elderly populations in these countries. In addition, the outcome of this study will also assist in executing focused health interventions for the most at-risk groups of the community, develop an environmental pollution monitoring system and strengthen environmental laws and regulations.

## 1.1. The concept of life expectancy and its determinants

Life expectancy is the average outstanding years of life at a specific age of an individual, which captures the prevailing patterns of mortality for various age groups [31] concluded that longer life expectancy is desirable for its inherent value as well as for the important life achievements of each individual. It is considered as one of the most critical parameters of the Human Development Index, and improvement of life expectancy is principal to much medical research. In addition, good health and longevity are related to higher productivity which is an essential stimulus for sustainable economic growth [15]. Income level is considered as one of the major drivers of life expectancy, and many researchers have concluded that higher income leads to greater life expectancy in a country [21, 32, 33]. For example, Mackenbach and Looman [34] found that rising national income reduced the mortality from infectious diseases in European countries over the period of 1990 to 2008 while they studied the upward shift of the Preston curve (the link between life expectancy and per capita real income) for the selected European countries. However, significant disparities in life expectancy are predominant among countries with identical per capita income [35]. For example, according to the World Bank [20] data, life expectancy in Bangladesh (72 years) and Nepal (70 years) are higher than in India (69 years) and South Africa (64 years), despite having lower per capita income [20].

Understanding the determinants of the life expectancy of a nation is a complex issue. Many lifestyles, nutritional, environmental, genetic and socioeconomic factors can affect people's health and longevity [36, 37]. Healthcare expenditure is also revealed as a factor with a strong positive impact on life expectancy in the studies of Bein et al. [38], Jaba et al. [39] and Ranabhat et al. [40]. In terms of developed countries [41–43] found that increasing health expenditure positively impacts life expectancy. In another study on 40 countries of sub-Saharan Africa (SSA), Arthur and Oaikhenan [44] also revealed the improved life expectancy due to increased healthcare expenditure. However, van del Heuvel and Olaroiu [45] and Rahman et al. [22] found no impact of healthcare expenditure on the life expectancy of 31 European countries and SAARC-ASEAN regions, respectively. The studies of Filmer [46] and Barlow and Vissand-jee [47] also support this no impact result.

Sanitation is also linked to life expectancy. Poor sanitation causes the transmission of many diseases such as cholera, diarrhea, hepatitis A, typhoid, etc., reducing life expectancy [48]. According to this report, around 432,000 deaths each year occur mainly due to poor sanitation. Similarly, unclean or contaminated drinking water transmits various diseases that adversely affect life expectancy via infant mortality [22, 49]. WHO Report [50] also notes 485 000 diarrheal deaths each year, mostly related to unclean drinking water. Islam et al. [51] used healthy life expectancy (HALE) data to evaluate the health status and quality of life in lower-middle and low income countries. Along with other known factors, they have found economic freedom, level of corruption, carbon dioxide emission and success in achieving millennium development goals are highly correlated to higher life expectancy.

Past empirical studies have identified other determinants of life expectancy such as lifestyle and occupation [52], nutrition and food availability [53], government expenditure on social protection and education level of the population [54], and availability of healthcare services and professionals [55] Auster et al. (1969) [56] examined the association between medical care and environmental variables with mortality in the USA. This seminal work concluded that environmental factors (e.g. education, income, diets, physical activities, and psychological health) were more important in reducing mortality in comparison to medical care. Recently, in a similar study Thornton, J. (2010) [39] found that death rates are related to socioeconomic status and lifestyle. The study suggested that medical care services are unable to improve the nation's health status significantly if a country ignores key policies that improve economic, social, and lifestyle factors.

The current study attempted to incorporate all the available variables (determining life expectancy) into the empirical model to identify the factors influencing life expectancy in the 31 most polluted countries in the world. However, some key variables such as education level and lifestyles were not available for all the countries for the period of 2000–2017.

## 2. Data and methods

### 2.1. Data

This study uses balanced panel data over the period of 2000–2017 for 31 world's most polluted countries. Most polluted countries are selected where the average PM2.5 (mg/m3), an air pollutant, is greater than 20, and these data are collected from World Population Review (WPR, 2020). The countries are Afghanistan, Bahrain, Bangladesh, Bulgaria, Cambodia, Chile, China, Croatia, Czech Republic, Ethiopia, India, Indonesia, Iran, Kazakhstan, Korea Republic, Kuwait, Mexico, Mongolia, Nepal, Nigeria, Pakistan, Peru, Poland, Serbia, Sri Lanka, Thailand, Turkey, Uganda, United Arab Emirates, Uzbekistan and Vietnam. Also see S2 Appendix.

The data were acquired from the World Development Indicator [20], World Bank open database. The carbon emissions data for the period from 2015 to 2017 are not available in the

**Table 1. Descriptive statistics of the variables.**

| Variables | Mean | Median | Standard Deviation | Minimum | Maximum |
|---|---|---|---|---|---|
| LIF (total years) | 70.39 | 72.75 | 7.17 | 46.23 | 82.63 |
| $CO_2$ (metric tons per capita) | 5.93 | 3.94 | 7.27 | 0.04 | 35.92 |
| GDP (per capita US$) | 8565.45 | 4188.70 | 11319.38 | 194.87 | 63251.52 |
| HEX (per capita US$) | 700.51 | 420.11 | 688.33 | 21.38 | 3070.09 |
| WAT (% of total population) | 85.27 | 92.29 | 18.86 | 18.70 | 100.00 |
| SAN (% of total population) | 71.25 | 85.89 | 29.47 | 3.40 | 100.00 |

WDI; therefore, these are sourced from the British Petroleum (BP) Statistical Review of World Energy [57]. The world's most 31 polluted countries are selected where average PM2.5 (mg/m3), an air pollutant, is greater than 20, and these data are collected from World Population Review [58]. Table 1 shows the summary statistics of the variables that are used in the study. The average life expectancy at birth is around 70 years, GDP per capita is $8,566, and per capita health expenditure is $701. On average, 85% of the population can use basic drinking water and 71% population use sanitation service. Average per capita $CO_2$ emissions are 6 metric tons in the sample countries.

## 2.2. Model

Preston [33] develops a model, known as Preston Curve, to explore the relationship between life expectancy and real GDP per capita and found a positive link between these two variables. The basic model of the Preston Curve is noted below:

$$LIF = f(GDP) \tag{1}$$

Where LIF and GDP represent life expectancy and real GDP per capita (a proxy for economic growth), respectively. The coefficient of GDP is expected to have a positive sign. This study uses the augmented model of Preston Curve by adding some other relevant explanatory variables as stated above. Therefore, the used model for the study is as follows:

$$LIF = f(GDP, \ CO_2, \ HEX, \ WAT, \ SAN) \tag{2}$$

$CO_2$ emissions are believed to impact human life expectancy [28, 59, 60] as a major determinant. It is expected that $CO_2$ emissions have a negative relationship with life expectancy. We expect a positive link between LIF and the rest of the explanatory variables. This study uses panel data so that our baseline model will be re-written as follows:

$$LIF_{it} = \beta_0 + \beta_1 GDP_{it} + \beta_2 CO_2 it + \beta_3 HEX_{it} + \beta_4 WAT_{it} + \beta_5 SAN_{it} + \varepsilon_{it} \tag{3}$$

Subscripts i and t indicate country and year, respectively. β1- β5 are the vectors of coefficients for time-varying explanatory variables. $\varepsilon_{it}$ is the error terms for country i at year t. All variables are transformed into natural logarithms in order to reduce heteroscedasticity.

$$lnLIF_{it} = \beta_0 + \beta_1 lnGDP_{it} + \beta_2 lnCO_{2it} + \beta_3 lnHEX_{it} + \beta_4 lnWAT_{it} + \beta_5 lnSAN_{it} + \varepsilon_{it} \tag{4}$$

## 2.3. Econometric approach

This research conducts a panel data approach as this analysis has certain advantages. First, it has both time-series and cross-sectional dimensions. Second, the panel data analysis addresses the individual heterogeneity issue. Third, this analysis reduces multi-collinearity and increases the degrees of freedom. Lastly, it overcomes the problems associated with time-series analysis [61].

**2.3.1. Panel unit root tests.** The test for panel unit root is the first necessary step to verify the stationary properties of the variables. A number of panel unit root tests exist in the literature. In this study, we use four first- and second-generation panel unit root tests for enhancing the robustness of results. They are Pesaran [62] test, Im, Pesaran and Shin (IPS) [63] test, Fisher [64] augmented Dickey–Fuller (ADF) test and Harris and Tzavalis [65] unit-root test. The null hypothesis for the panel unit root tests is: each data series is non-stationary at the level but stationary at the first difference across countries. The formulas for the various tests are shown in S3 Appendix.

**2.3.2. Cross-sectional dependence, autocorrelation and heteroscedasticity.** Panel data with autocorrelation, cross-sectional dependence and heteroscedasticity make serious problems for econometric analysis. The existence of cross-sectional dependence in a panel study indicates that there exists a common unnoticed shock among the cross-sectional variable over a time period [66].

Khan et al. [67] define autocorrelation as "the disturbance term correlated with any variable of the model that has not been affected by the disturbance term related to other variables in this model." Heteroscedasticity arises when the variance of the disturbance differs across samples [68].

Parks [69] proposes Feasible Generalized Least Squares (FGLS), which is efficient in overcoming group-wise heteroscedasticity, time-invariant cross-sectional dependence and serial correlations. Beck and Katz [70] suggest an alternative panel-corrected standard error (PCSE) estimates to deal with the panel nature of the data. It is believed that FGLS and PCSE effectively deal with heteroscedasticity, serial correlations and cross-sectional dependence. Le and Nguyen [71] advocate that PCSE and FGLS are two techniques that rectify for autocorrelation and heterogeneity and yield robust standard errors. Ikpesu et al. [72] incorporate the PCSE approach to address autocorrelation, correct standard error estimate and overcome outlier estimates. Some previous studies use FGLS, which overcomes heteroscedasticity and autocorrelation [73, 74]. Alonso et al. [75] use PCSE and FGLS estimates for their panel data set and report similar results.

This study uses the time-series-cross-sectional Prais-Winsten (PW) regression with panel-corrected standard errors (PCSE) as a baseline estimate, which allow for disturbances that are contemporaneously correlated and heteroskedastic across the panel. The PCSE correction facilitates in avoiding statistical overconfidence, which is often connected with the feasible generalized least-square estimator where the total periods are smaller than total sample countries [70, 76].

## 3. Results

This study sample consists of 31 countries, and the period of study is for 18 years, 2000–2017. First, this study tests for the existence of heteroscedasticity, cross-sectional dependence and autocorrelation. Also, to investigate the stationary of the variables, this study adopts the Pesaran [62] CIPS, the Im-Persaran-Shin unit root test [63] and the Levin-Lin-Chu unit root test [77].

Table 2 shows that the cross-sectional dependence exits in all of the variables which can arise because of spatial or spill over effects or due to unobserved common factors [78]. Due to the presence of cross-section dependence, both the standard homogeneous estimators for panel data (Fixed-effect, Random-effect, or First Difference) and the heterogeneous Mean Group estimator are inconsistent [79]. Hence, we addressed this issue to avoid significant size distortion in the regression analysis. Besides, most of the variables are stationary at the levels,

**Table 2. The results of cross-sectional dependence and stationary test.**

|  | CD test | Pesaran (2007) CIPS | Im-Pesaran-Shin unit-root test | | Fisher unit root test | | Harris-Tzavalis unit-root test | |
|---|---|---|---|---|---|---|---|---|
|  | Statistics | Statistics | Statistics | *p*-value | Statistics | *p*-value | Statistics | *p*-value |
| lnLIF | 84.872*** | -2.521*** | -2.983*** | 0.001 | 213.070*** | 0.000 | 0.955 | 0.999 |
| lnCO₂ | 18.237*** | -2.438*** | 2.055 | 0.980 | 123.292*** | 0.000 | 0.908 | 0.988 |
| lnGDP | 62.742*** | -2.027 | 5.207 | 1.000 | 122.378*** | 0.000 | 0.962 | 1.000 |
| lnHEX | 77.525*** | -2.201* | -1.774** | 0.038 | 186.735*** | 0.000 | 0.787** | 0.029 |
| lnWAT | 58.409*** | -1.723 | -9.732*** | 0.000 | 170.471*** | 0.000 | 0.968 | 1.000 |
| lnSAN | 68.482*** | -2.312** | -13.904*** | 0.000 | 227.404*** | 0.000 | 0.9246 | 0.998 |

Note

***, **, and * indicate significance level at 1%, 5% and 10%, respectively.

which indicated that the individual observed series are stationary around a deterministic level [80] and the fixed, random effect and pooled OLS models are fit for this study [81].

Table 3 demonstrates the results of heteroscedasticity and autocorrelation, indicating that heteroscedasticity and auto-correlation exist in our used panel data. In this context, this study adopts the Panel-Corrected Standard errors model (PCSE) to explore the long-run effects of carbon emissions on life expectancy following the panel data estimation, as shown in Eq 4. This method has been adopted following Bailey and Katz [82], Jönsson [83], Le et al. [84], and Marques and Fuinhas [85] to address the heteroscedasticity, cross-sectional dependence and auto-correlation of variables in a small data with a short period (T) and large cross-sectionals (N). Following the previous studies, this study also uses the FGLS method for checking the robustness of results [84, 86–88]. Following Asongu et al. [89] and Bergh and Nilsson [90], this study also uses the fixed effect regreions that adjust for clustering over countries as a complementary analysis because it can correct within panel heteroscedasticity and autocorrelation.

Table 4 reports the PCSE long-run estimation results concerning the impact of life expectancy for 31 most polluted countries over the period 2000–2017. As expected, a 1% increase in per capita GDP and health expenditure increases the life expectancy by 0.013% and 0.024%, respectively. Carbon emissions have a significantly negative impact on life expectancy, suggesting that higher the carbon emissions lower the life expectancy. More specifically, a 1% increase of carbon emissions, keeping all other variables constant, decreases life expectancy by 0.012%. Therefore, this study finds that carbon emissions is a vital driver of life expectancy. Drinking water and sanitation have significantly positive impacts on life expectancy as well, and the effect of access to drinking water is substantial implying that 1% increase of this variable increases the life expectancy by 0.21%.

For robustness checks, this study also estimates a model using FGLS. Table 5 reports the determinants of life expectancy. Economic growth appears to have significantly positive effects on life expectancy supporting Preston Curve. The carbon emissions are shown to have negative effects on life expectancy; health care expenditure, water and sanitation appear to have significant and positive effects on life expectancy. Overall, the results from FGLS demonstrate consistent results with PCSE estimates.

**Table 3. The results of heteroscedasticity and autocorrelation.**

| Test | Test statistic | *p*-value | Decision |
|---|---|---|---|
| Modified Wald test for groupwise heteroskedasticity | X² = 44115.37 | 0.0000 | There is heteroscedasticity in the panel |
| Wooldridge test for autocorrelation in panel data | F-statistic = 84.29 | 0.0000 | The autocorrelation is present in the panel. |

**Table 4. The results of PCSE regression.**

|  | PCSE |
|---|---|
| _Constant | 3.006 (125.00)*** |
| lnGDP | 0.013 (4.39)*** |
| lnCO$_2$ | -0.011 (-7.83)*** |
| lnHEX | 0.024 (6.07)*** |
| lnWAT | 0.206(37.61)*** |
| lnSAN | 0.020 (4.62)*** |
| R-squared | 0.999 |
| Wald chi$^2$ | 5186.23 |
| Probability | 0.000 |
| N | 558 |

Note

*** denotes significance at 1% level. Figures in the parentheses are z-statistics.

To address the impact of the time trend in the panel data model, this study re-estimated the PCSE and FGLS model using a time trend variable using the assumption of a linear trend in the outcome variables over time. The results demonstrated identical coefficient sings which re-established the soundness of the econometric analysis. The findings are noted in Table 6 and Table 7 in S1 Appendix.

## 3.1. The results of causality test

Table 6 shows the short-term causality between life expectancy, carbon emissions, economic growth, healthcare expenditure, drinking water and sanitation. This study finds that there is a one-way causality running from carbon emissions to life expectancy. In other words, more carbon emissions threaten life expectancy. Additionally, this study reveals that there are bidirectional causal links between life expectancy and drinking water as well as life expectancy and sanitation. The study, however, found no short-run causality between GDP and life expectancy and between health expenditure and life expectancy.

It is worthy to note that there are some limitations that we faced in terms of data, variable selection, statistical measurements, and estimated results. First, this study had to select a short period of the data set (2000–2017) just because data for all selected variables for all countries

**Table 5. Robustness check: The results of FGLS regression.**

|  | FGLS |
|---|---|
| Constant | 3.105 (79.19)*** |
| lnGDP | 0.022 (10.49)*** |
| lnCO$_2$ | -0.011 (-7.97)*** |
| lnHEX | 0.012 (6.57)*** |
| lnWAT | 0.162 (15.29)*** |
| lnSAN | 0.042 (9.78)*** |
| Wald chi$^2$ | 2866.08 |
| Probability | 0.000 |
| N | 558 |

Note

*** denotes significance at 1% level. Figures in the parentheses are z-statistics.

**Table 6. Pairwise granger causality tests.**

| Null Hypothesis: | F-Statistic | Causality |
|---|---|---|
| $lnCO_2 \rightarrow lnLIF$ | 4.27** | One-way causality from $lnCO_2$ to $lnLIF$ |
| $lnLIF \rightarrow lnCO_2$ | 2.13 | |
| $lnGDP \rightarrow lnLIF$ | 0.32 | No causality between $lnGDP$ and $lnLIF$ |
| $lnLIF \rightarrow lnGDP$ | 2.19 | |
| $lnHEX \rightarrow lnLIF$ | 0.08 | No causality between $lnHEX$ and $lnLIF$ |
| $lnLIF \rightarrow lnHEX$ | 1.58 | |
| $lnWAT \rightarrow lnLIF$ | 21.48*** | Two-way causality between $lnWAT$ and $lnLIF$ |
| $lnLIF \rightarrow lnWAT$ | 8.87*** | |
| $lnSAN \rightarrow lnLIF$ | 11.26*** | Two-way causality between $lnSAT$ and $lnLIF$ |
| $lnLIF \rightarrow lnSAN$ | 11.94*** | |

Note

***, **, and * indicate significance level at 1%, 5% and 10%, respectively.

were not available beyond this period when the study was conducted. Since this study is based on balanced panel data set, consideration of extended period was not possible. Second, the estimation used two data sources: World Bank and BP statistics, because $CO_2$ emissions data were not available in the World Bank data source for the last three years (2015–2017). Third, the study could not use some other possible variables like lifestyle factors (smoking/drinking habit, physical exercise), government policies, literacy rates, physician/people ratio, etc. due to the paucity of data. These variables may also affect life expectancy. Fourth, cross-sectional dependence, heteroscedasticity and autocorrelation were found in the panel data. To address this last limitation, this study used appropriate estimation methods, PCSE and FGLS regressions.

# 4. Discussions

This paper investigates the determinants of life expectancy in 31 most pullulated countries of the world with a special focus on environmental degradation (measured by $CO_2$ emissions). These countries are also low-middle income countries. Taking the BP and World Bank annual data for the period of 18 years (2000–2017), we have used the PCSE model to estimate the long run effects of environmental degradation on life expectancy. Then we have applied FGLS regression to check the consistency of the results found in PCSE regression. We also check the cross-sectional dependence and perform other essential diagnostic tests for panel data. The results from both PCSE and FGLS regressions confirm a significant negative effect of $CO_2$ emissions on life expectancy, whist all other variables (GDP per capita, health expenditure per capita, people's access to basic drinking water services and improved sanitation services) are positively correlated with life expectancy. The Pairwise Granger Causality Tests show one-way causal link from carbon emissions to life expectancy and bidirectional causal links between life expectancy and drinking water, and life expectancy and sanitation. Thus, our results identify that environmental degradation is a threat for attaining the improved life expectancy in the sample countries.

Our findings showed that economic growth has a significant positive association with life expectancy, supporting Preston Curve. This means that higher economic growth would most likely increase the life expectancy of people living in the world's most polluted countries. This finding is consistent with theory and the previous research evidence (see Luo and Xie (2020) [91] and Wang et al. (2020) [92]) that higher economic growth increases more years for life

expectancy. There are several reasons. Previous studies have shown that increasing national income reduces the adverse impact of infectious diseases in the communities Mackenbach and Looman [34], increases food availability and consumption [53], and government expenditure on social protection [54]. Increasing income is also associated with the higher education level of the population [54]. Therefore, increased income is one of the major factors determining life expectancy in polluted countries.

The healthcare expenditure has a significant positive impact on life expectancy, implying that higher healthcare expenditure would increase life expectancy. This result is in line with the results of previous studies indicating that healthcare expenditure is an important factor in life expectancy Bein et al. [38], Jaba et al. [39] and Ranabhat et al. [40, 41, 93]. In addition, higher health expenditure is associated with greater availability of healthcare services and professionals [94]. Increased availability might have increased the access and use of healthcare in these 31 countries. Hence, we have found a positive impact of health expenditure on life expectancy.

This study also found that increasing access to clean water and improved sanitation improves life expectancy. These results are similar to previous findings of [22, 49]. Improved water and sanitation quality reduce waterborne diseases (particularly in lower-income countries), reducing mortality rates.

Finally, our study showed that increasing carbon emissions negatively impacts life expectancy (holding other variables constant) in the 31 most polluted countries in the world. Previously, numerous studies have found similar associations in developing and developed countries (see Pope III [7, 95], World Health Organization [3]). Although the current study is the first to examine the carbon emmissions and life expectancy nexus for most polluted countries, the similarity with past empirical findings is justifiable. For example, Apergis et al. [5] and Kampa [6] showed that outdoor air pollution causes severe chronic diseases that increase mortality. Furthermore, Wen and Gu [12] and Wang et al. [13] concluded that air quality adversely impacts the longevity of the older population, particularly those suffering from various comorbidities. Majeed and Ozturk [35] associated air pollution with higher levels of infant mortality and Pope III [28] estimated a 15% increase in life expectancy due to a reduction in air pollution in the United States during the 1980s and 1990s. Therefore, we believe with the reduction in air pollution; the selected 31 most polluted countries could improve the life expectancy of the population to a significant level.

Based on our findings, several policy recommendations can be drawn. First, the policy makers should implement strong environmental policies that reduce pressure on environmental resources such as water, land, forest and air quality. Evidently, the most polluted countries feature the weakest environmental policies, and they often fail to implement public policies to downgrade environmental damages caused by rapid economic growth. Since, environmental pollution results in a poor quality of life, it often impedes the positive impact of economic growth on life expectancy [96]. Numerous past studies have concluded that healthier nations have higher per capita productivity and are able to accumulate more wealth compared to those with poor health [97, 98]. Therefore, policy marker of these countries should adopt effective public health and environmental policies that will pay off in the long run in terms of better health from reduced $CO_2$ emission and thus increases productivity and economic growth. They should also invest in research and innovation to invent and produce technologies that will reduce environmental degradation in addition to developing an environmental pollution monitoring system and strengthening environmental laws and regulations. Second, production activities for higher economic growth should continue using environment-friendly technologies and resources such as renewable energy. Third, since growth in income and health expenditure have positive effects on the life expectancy, budgetary allocation on health care

expenditure must be increased. Finally, basic sanitation facilities and clean drinking water for all must be ensured to improve the life expectancy in these countries. The joint efforts through public-private initiatives will be helpful in this regard.

This study the first attempt to evaluate the association between pollution and life expectancy for the 31 most polluted countries in the world. The findings would provide policymakers of these countries to re-evaluate their environmental policies and practices. The results should also assist them in making strong arguments for air quality improvements. The findings also present a strong case for investing in safe drinking water and sanitation facilities in these countries.

## 5. Conclusions

Overall, this study used the latest and sophisticated econometric techniques to estimate the determinants of life expectancy for the most polluted countries in the world. In this context, carbon emission was confirmed as the key determining factor. For these 31 countries, rising $CO_2$ emissions had a significant negative impact on life expectancy both in short as well as in the long-run. Variables such as the availability of safe drinking water, and improved sanitation facility, increased the life expectancy. Although rising GDP and expenditure on health promote higher life expectancy, this study did not find a short-run causal relationship from the direction of GDP to life expectancy or health expenditure to life expectancy. This would suggest that countries with very high pollution level may not achieve a higher life expectancy in the short-run, despite having positive GDP growth and expanding healthcare expenditure.

This study has some limitations which could be addressed in future analysis. Firstly, due to unavailability of the data, key variables that determine life expectancy such as education level of the population, income inequality, diseases burden, nutrition and diet, and lifestyles were not included in the estimated model. Secondly, although we have used $CO_2$ emissions (metric tons per capita) as a measure of pollution, there are other common measure of pollution such as $PM_{2.5}$ or $PM_{10}$ (fine particular matter) concentrations. Thirdly, this study did not control for individual risk factors (e.g. obesity, smoking habit or alcohol consumption) that might impact life expectancy. Lastly, there is a high probability that the negative impact of pollution could be different among countries due to income level or access to and availability of healthcare services. Hence, it is unclear if the findings are generalizable outside these 31 most polluted countries. Future studies should address these issue. Furthermore, it is also important to understand whether the negative impact of pollution is more prominent on people from lower socioeconomic background, people with occupational exposure, older age, and people with comorbidities. More comprehensive analysis and understanding of the adverse impact of pollution on life expectancy is required.

## Supporting information

**S1 Appendix.**
(DOCX)

**S2 Appendix.**
(DOCX)

**S3 Appendix.**
(DOCX)

## Author Contributions

**Data curation:** Mohammad Mafizur Rahman.

**Formal analysis:** Mohammad Mafizur Rahman.

**Methodology:** Mohammad Mafizur Rahman, Rezwanul Rana.

**Supervision:** Rasheda Khanam.

**Visualization:** Rasheda Khanam.

**Writing – original draft:** Mohammad Mafizur Rahman, Rezwanul Rana.

**Writing – review & editing:** Mohammad Mafizur Rahman.

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
