## [Decision Letter · Decision Letter 0]

10 Nov 2021

PONE-D-21-32920Determinants of Life Expectancy in Most Polluted Countries: Exploring the Effect of Environmental DegradationPLOS ONE

Dear Dr. Rahman,

Thank you for submitting your manuscript to PLOS ONE. After careful consideration, we feel that it has merit but does not fully meet PLOS ONE’s publication criteria as it currently stands. Therefore, we invite you to submit a revised version of the manuscript that addresses the points raised during the review process.

We look forward to receiving your revised manuscript.

Kind regards,

María del Carmen Valls Martínez, Ph.D.

Academic Editor

PLOS ONE

Journal Requirements:

"This research article has received no fund from any source. Authors have dedicated their time for the benefit of policy makers, other researchers and general public/readers.

Reviewers' comments:

Reviewer's Responses to Questions

**Comments to the Author**

1. Is the manuscript technically sound, and do the data support the conclusions?

Reviewer #1: Partly

Reviewer #2: Partly

2. Has the statistical analysis been performed appropriately and rigorously? 

Reviewer #1: Yes

Reviewer #2: Yes

3. Have the authors made all data underlying the findings in their manuscript fully available?

Reviewer #1: Yes

Reviewer #2: No

4. Is the manuscript presented in an intelligible fashion and written in standard English?

Reviewer #1: Yes

Reviewer #2: Yes

5. Review Comments to the Author

Reviewer #1: First of all, I would like to thank the editorial board of Plos One for their confidence in my tasks as a reviewer. Regarding the review of the paper "Determinants of Life Expectancy in Most Polluted Countries: Exploring the Effect of Environmental Degradation", Manuscript Number: PONE-D-21-32920, the following is the outcome of my review:

Major Revision

In my opinion, this is not a bad paper at all. However, there are a number of weaknesses that do not recommend its acceptance in its current state. In this regard, I will indicate below a series of changes and proposals for improvement that I recommend that should be carried out with a view to a plausible acceptance of the article.

- Background section. I consider that this work lacks an introductory section that begins the work, stating its objectives, achievements and scope, ending with a brief paragraph indicating the topics to be developed in each of the subsequent subsections and the hypothesis to be verified.

- It is also necessary for the background section to include a more elaborate state of the question than the one presented in this paper in order to contextualize and conceptualize the term "Life Expectancy" in a balanced way. On the other hand, this work omits some previous articles, some of them already classics such as the ones I indicate below, which I strongly suggest to include as a bibliographic basis for this paper:

Translated with www.DeepL.com/Translator (free version)

o Auster, R., Levesoardln, I. and Sarachek, S. (1969), 'The production of health: An exploratory study', J. Hum. Resour 4, 411-436.

o Crémieux, P.-Y., Ouellette, P. and Pilon, C. (1999), 'Health care spending as determinants of health outcomes', Health Econ 8, 627-639.

o Crémieux, P., Mieilleur, M., Ouellette, P., Petit, P., Zelder, P. and Potvin, K. (2005), 'Public and private pharmaceutical spending as determinants of health outcomes in Canada', Health Econ 14, 107-116.

o Halicioglu, F. (2011), 'Modeling life expectancy in Turkey', Econ. Model 28, 2075-2082.

o Hitiris, T. and Posnett, J. (1992), 'The determinants and effects of health expenditure in developed countries', J. Health Econ 11, 173-181.

o Martín Cervantes, P. A., Rueda López, N. and Cruz Rambaud, S. (2019), 'A Causal Analysis of Life Expectancy at Birth. Evidence from Spain', International Journal of Environmental Research and Public Health 16(13), 2367.

o Martín Cervantes, P. A., Rueda López, N. and Cruz Rambaud, S. (2020), 'The Relative Importance of Globalization and Public Expenditure on Life Expectancy in Europe: An Approach Based on MARS Methodology', International Journal of Environmental Research and Public Health 17(22), e8614.

o Thornton, J. (2002), 'Estimating a health production function for the US: Some new evidence', Appl. Econ 34, 59-62.

o Wolfe, B. and Gabay, M. (1987), 'Health status and medical expenditures: More evidence of a link', Soc. Sci. Med 25, 883-888.

- Data

Please create a table with the 31 countries selected in your study and indicate the following, ordered from highest to lowest: Country-Pollution degree-Life expectancy years.

On the other hand, what is the reason for choosing "PM2.5 (mg/m3), an air pollutant, is greater than 20? Was this choice made on the basis of any previous work? Does it find any support in the literature or is it a criterion freely used by the authors? Please specify.

Regarding the table with descriptive statistics, perfect, but make a minimum comment on these statistics.

- Panel data unit tests

To be frank, I do not see any logic or usefulness in developing the formulas for the various tests used. In a textbook probably yes, but not in a scientific article. Therefore, I recommend eliminating such development or moving it to an appendix section.

In the same way, I do consider it necessary to additionally perform the KPSS test (Kwiatkowski-Phillips-Schmidt-Shin (KPSS)) since it would supplement your results from an alternative point of view, taking into account that contrary to most unit root tests, the presence of a unit root is per se not the null hypothesis but the alternative hypothesis.

- Please include the Granger causality test among the citations used and, of course, include the limitations found in the conclusions.

- The results obtained are quite good and congruent. It can be seen that the 31 countries selected are low-middle income countries. Please note this fact and, of course, include it in the discussion section. On the other hand, the discussion section is excessively sparse; please explain it with much more detail. The results obtained in the light of the literature should be presented in much greater depth, using also the suggested papers and the new section devoted to the literature review that you have to prepare.

Once you make each and every one of the suggested changes, I would be delighted if this work is finally accepted.

With my best wishes in your personal and academic life,

The reviewer

Reviewer #2: I found the article very interesting and very well done from a methodological point of view. However, in order to be published, it requires a series of adjustments in its structure:

- The first section should be called Introduction and should be the introduction, not the literature review. This section should include all the aspects required for an introduction. Such as, state of the art, summary of results, GAP, contributions, etc.

- It should have a subsequent section called literature review and hypothesis setting.

- The results section should be only the presentation of the results and not a discussion there. This happens for example in line 336.

-The article does not have a discussion section as required. In the one included by the authors there is no discussion of the results obtained and contrast with those of authors of previous research.

-The conclusion section is very weak. It is necessary to indicate the main contributions of the research, as well as the possible limitations to the scope and future lines of research arising from this article.

In view of the above, I consider that the article requires major changes in order to be published.

6. PLOS authors have the option to publish the peer review history of their article (what does this mean?). If published, this will include your full peer review and any attached files.

Reviewer #1: No

Reviewer #2: No

---

## [Author Response · Author response to Decision Letter 0]

6 Dec 2021

Revised version with response list has been updated.

---

## [Decision Letter · Decision Letter 1]

17 Dec 2021

PONE-D-21-32920R1Determinants of Life Expectancy in Most Polluted Countries: Exploring the Effect of Environmental DegradationPLOS ONE

Dear Dr. Rahman,

Thank you for submitting your manuscript to PLOS ONE. After careful consideration, we feel that it has merit but does not fully meet PLOS ONE’s publication criteria as it currently stands. Therefore, we invite you to submit a revised version of the manuscript that addresses the points raised during the review process.

Please submit your revised manuscript by Jan 31 2022 11:59PM If you will need more time than this to complete your revisions, please reply to this message or contact the journal office at plosone@plos.org. Please include the following items when submitting your revised manuscript:A rebuttal letter that responds to each point raised by the academic editor and reviewer(s). You should upload this letter as a separate file labeled 'Response to Reviewers'.A marked-up copy of your manuscript that highlights changes made to the original version. You should upload this as a separate file labeled 'Revised Manuscript with Track Changes'.An unmarked version of your revised paper without tracked changes. You should upload this as a separate file labeled 'Manuscript'.If applicable, we recommend that you deposit your laboratory protocols in protocols.io to enhance the reproducibility of your results. Protocols.io assigns your protocol its own identifier (DOI) so that it can be cited independently in the future. For instructions see: https://journals.plos.org/plosone/s/submission-guidelines#loc-laboratory-protocols. Additionally, PLOS ONE offers an option for publishing peer-reviewed Lab Protocol articles, which describe protocols hosted on protocols.io. Read more information on sharing protocols at https://plos.org/protocols?utm_medium=editorial-email&utm_source=authorletters&utm_campaign=protocols.

We look forward to receiving your revised manuscript.

Kind regards,

María del Carmen Valls Martínez, Ph.D.

Academic Editor

PLOS ONE

Journal Requirements:

Additional Editor Comments (if provided):

The paper has been substantially improved after the revision. However, the final considerations made by reviewer 1 should be taken into account.

Reviewers' comments:

Reviewer's Responses to Questions

**Comments to the Author**

1. If the authors have adequately addressed your comments raised in a previous round of review and you feel that this manuscript is now acceptable for publication, you may indicate that here to bypass the “Comments to the Author” section, enter your conflict of interest statement in the “Confidential to Editor” section, and submit your "Accept" recommendation.

Reviewer #1: All comments have been addressed

Reviewer #2: All comments have been addressed

2. Is the manuscript technically sound, and do the data support the conclusions?

Reviewer #1: Yes

Reviewer #2: Yes

3. Has the statistical analysis been performed appropriately and rigorously? 

Reviewer #1: Yes

Reviewer #2: Yes

4. Have the authors made all data underlying the findings in their manuscript fully available?

Reviewer #1: No

Reviewer #2: Yes

5. Is the manuscript presented in an intelligible fashion and written in standard English?

Reviewer #1: Yes

Reviewer #2: Yes

6. Review Comments to the Author

Reviewer #1: Once again, I would like to thank the editorial board of Plos One for allowing me to review for such a renowned scientific publication. Focusing on my revision of the article (PONE-D-21-32920R1), below is my decision:

Accept

I must recognize the effort made by the authors to rework this paper, which I consider has gained a lot of added value after implementing the changes and proposals for improvement suggested by the reviewers, therefore, congratulations to the authors for the work done.

In any case, I would ask the authors to take into account the following points.

1. Use the "p" of the p-values always in italics.

2. Check if there is an error in Table 2 (Z-t-tilde-bar????).

3. It seems that the new references used do not appear in the list of final references. Please, check it.

4. The answer given for not using the KPSS test, believe me, is not valid at all. Keep this point in mind for future scientific works. Likewise, I suggest on a personal level that you do not support your views exclusively on Wikipedia.

5. I would consider that a greater emphasis on characterizing why in several non highly industrialized countries, which in many cases are geographically close, such high episodes of environmental pollution occur would have been mandatory.

Please bear in mind the points I have just made. In any case, congratulations.

With my best wishes in your personal and academic life,

The reviewer

Reviewer #2: I was pleased to see that the authors have taken my recommendations into account.

This has allowed the article to improve significantly compared to the first evaluated version.

For this reason, I consider the article suitable for publication.

7. PLOS authors have the option to publish the peer review history of their article (what does this mean?). If published, this will include your full peer review and any attached files.

Reviewer #1: No

Reviewer #2: No

---

## [Author Response · Author response to Decision Letter 1]

3 Jan 2022

Please see the attached response letter for the Editor and Reviewers.

---

## [Editor Report · Decision Letter 2]

6 Jan 2022

Determinants of Life Expectancy in Most Polluted Countries: Exploring the Effect of Environmental Degradation

PONE-D-21-32920R2

Dear Dr. Mohammad Mafizur Mafizur Rahman,

We’re pleased to inform you that your manuscript has been judged scientifically suitable for publication and will be formally accepted for publication once it meets all outstanding technical requirements.

Kind regards,

María del Carmen Valls Martínez, Ph.D.

Academic Editor

PLOS ONE

---

## [Editor Report · Acceptance letter]

11 Jan 2022

PONE-D-21-32920R2 

Determinants of Life Expectancy in Most Polluted Countries: Exploring the Effect of Environmental Degradation 

Dear Dr. Rahman:

I'm pleased to inform you that your manuscript has been deemed suitable for publication in PLOS ONE. Congratulations! Your manuscript is now with our production department. 

Kind regards, 

on behalf of

Dr. María del Carmen Valls Martínez 

Academic Editor

PLOS ONE